# Impact of the COVID-19 pandemic and typhoid conjugate vaccine introduction on typhoid fever in Nepal

Dipesh Tamrakar[1,2], Shiva Ram Naga[1], Esther Jung [3], Basudha Shrestha[4], Pratibha Bista Roka[5], Rabin Pokharel[6], Sabin Bikram Shahi[1], Aarjya Tara Bajracharya[1], Surendra K. Mahadup[7], Nishan Katuwal[1], Kate Doyle[8], Jessica C. Seidman[8], Alice S. Carter[8], Stephen P. Luby[3], Isaac I. Bogoch[9], Kristen Aiemjoy[10,11], Denise O. Garrett[8], Rajeev Shrestha[1,12], Jason R. Andrews [3]*

**1** Center for Infectious Disease Research and Surveillance, Dhulikhel Hospital Kathmandu University Hospital, Dhulikhel, Nepal, **2** Department of Community Medicine, Kathmandu University School of Medical Sciences, Dhulikhel, Nepal, **3** Division of Infectious Diseases and Geographic Medicine, Stanford University School of Medicine, Stanford, California, United States of America, **4** Microbiology Unit, Department of Pathology, Kathmandu Model Hospital, Kathmandu, Nepal, **5** Department of Pathology, Bir Hospital, Kathmandu, Nepal, **6** Helping Hands Community Hospital, Kathmandu, Nepal, **7** Department of Microbiology, Kathmandu University School of Medical Sciences, Dhulikhel, Nepal, **8** Sabin Vaccine Institute, Washington, DC, United States of America, **9** Divisions of Infectious Diseases and General Internal Medicine, Toronto General Hospital, University Health Network, Toronto, Ontario, Canada, **10** Division of Epidemiology, Department of Public Health Sciences, University of California Davis School of Medicine, Sacramento, California, United States of America, **11** Department of Microbiology and Immunology, Mahidol University Faculty of Tropical Medicine, Bangkok, Thailand, **12** Department of Pharmacology, Kathmandu University School of Medical Sciences, Dhulikhel, Nepal

\* jandr@stanford.edu

## Abstract

### Background

While typhoid conjugate vaccines (TCV) offer promise for reducing risk in endemic settings, their population-level impact remains unclear. In 2022, Nepal introduced TCV nationally on the heels of the COVID-19 pandemic, which disrupted healthcare services, surveillance, and potentially typhoid transmission dynamics, complicating vaccine impact evaluation. We investigated the impact of TCV introduction amid shifting typhoid burden during the pandemic.

### Methods

We analyzed blood culture data from four Kathmandu Valley health facilities, comparing culture positivity for *Salmonella* Typhi across three periods: pre-pandemic (January 2018-March 2020); pandemic, pre-vaccine introduction (April 2020-March 2022); post-vaccine introduction (April 2022-April 2024). We used multivariable logistic regression to assess *S.* Typhi positivity, adjusting for month and site, stratified by TCV-eligible children and older, TCV-ineligible populations.

**Data availability statement:** Data used to generate the figures and conduct descriptive analyses are publicly available in the Dryad Digital Repository at https://datadryad.org/dataset/doi:10.5061/dryad.kprr4xhhj . Additional individual-level data underlying the analyses are being aggregated and is available at the same DOI.

**Funding:** This study was funded by the Gates Foundation (IV-008335 to DOG). The funder had no role in study design, data collection and analysis, decision to publish, or preparation of the manuscript.

**Competing interests:** We have read the journal's policy and the authors of this manuscript have the following competing interests: IIB has consulted to the Weapons Threat Reduction Program at Global Affairs Canada. All other authors declare that no competing interests exist.

## Results

Between January 2018 and April 2024, 62,236 blood cultures were performed. *S.* Typhi blood culture positivity decreased from 2.11% pre-pandemic to 0.59% during the pandemic (p < 0.001) and remained low at 0.69% after TCV introduction. Among TCV-eligible children (15 months to 15 years), odds of *S.* Typhi positivity during the pandemic were 47% lower than the pre-COVID period (aOR 0.53, 95% CI 0.29-0.90) and continued to decrease by 75% post-TCV introduction (aOR 0.25, 95% CI 0.11-0.55). In contrast, among vaccine-ineligible individuals (≥16 years), odds of positivity during the pandemic were 77% lower than the pre-COVID period (aOR 0.23, 95% CI 0.16-0.31) but increased by 59% following TCV rollout (aOR 1.59, 95% CI 1.14-2.27). Sensitivity analyses restricted to pathogen-positive cultures yielded similar results.

## Conclusion

*S.* Typhi blood culture positivity declined sharply during the pandemic before TCV introduction. The subsequent rollout of TCV substantially reduced typhoid burden in vaccine-eligible children; however, rising cases among older, vaccine-ineligible populations following the relaxation of pandemic measures highlights the need for additional control measures such as improved water and sanitation infrastructure and broader age eligibility for typhoid vaccination.

### Author summary

Typhoid fever is a serious illness caused by bacteria that spread through contaminated food and water. In 2022, Nepal introduced a new typhoid vaccine for children, just as the country was recovering from the COVID-19 pandemic. The pandemic transformed how people accessed healthcare and may have also affected how typhoid spread. We wanted to understand how both the pandemic and the vaccine rollout influenced typhoid infections over time. Using results from four hospitals in the Kathmandu Valley between 2018 and 2024, we compared the number of typhoid cases before the pandemic, during the pandemic before typhoid vaccine introduction, and after typhoid vaccine introduction. We found that typhoid cases dropped sharply during the pandemic, even before the typhoid vaccine introduced. After the vaccine was introduced, typhoid continued to decline in children who were eligible for the vaccine. However, cases increased in people who were not vaccinated. Our findings show that the vaccine helped protect the age group that received it, but highlights the need for additional interventions such as improvements to sanitation and clean water infrastructure, and expanding vaccine access to more age groups to control typhoid in the future.

## Introduction

Typhoid is a life-threatening systemic infection caused by Salmonella enterica serovar Typhi (S. Typhi), which causes an estimated 11–21 million illnesses and 135,000–200,000 deaths annually in low- and middle-income countries (LMICs) [1–3]. The majority of cases and deaths occur in LMIC settings with limited access to clean water, poor sanitation, and overcrowded living conditions. In 2018, the World Health Organization recommended that countries with a high burden of typhoid introduce typhoid conjugate vaccines (TCVs) [4], which have demonstrated high efficacy in randomized trials [5–7]. Several countries have now introduced typhoid conjugate vaccines (TCVs) into national immunization programs. However, the population level impact of such introductions is not yet clear.

Nepal is a high typhoid burden country, with estimated incidence between 250 and 1,000 cases annually per 100,000 persons [8,9]. One of the pivotal Phase 3 trials assessing the effectiveness of TCVs was completed in Lalitpur, in the Kathmandu Valley [5] and on April 13, 2022, Nepal became the fourth country in the world to introduce TCVs nationally. The TCV roll-out began with a campaign targeting children aged 15 months to 15 years and was followed by introduction into the routine immunization program alongside the second dose measles and rubella (MR) vaccine at 15 months [10]. The catch-up campaign was estimated to have achieved greater than 90% coverage nationally [11].

In the two years prior to Nepal's introduction of TCV, the COVID-19 pandemic impacted the country's health services and disease surveillance systems. Mitigation measures including lockdowns, control of public gatherings, and closures of educational institutions and public spaces (restaurants, malls, etc.) may have altered transmission patterns of many pathogens beyond SARS-CoV-2, including typhoidal *Salmonella*. Reductions in enteric disease notifications during the COVID-19 pandemic have been reported in several countries [12–19]. Distinguishing reductions in infections from declines in notifications due to altered health care-seeking behaviors or weakening of surveillance systems can be difficult. In this context, understanding the impact of TCV introduction on typhoid incidence requires careful analysis.

In Nepal, the Surveillance for Enteric Fever in Asia Project (SEAP) performed systematic surveillance for typhoid at multiple hospitals in and around the Kathmandu Valley from 2016 – 2024 [8]. We leveraged this system to investigate the impact of TCV introduction amid changing typhoid burden during the COVID-19 pandemic. This study provides a unique opportunity to assess typhoid trends over a seven-year period encompassing the pre-pandemic, pandemic, and post-vaccine introduction phases, using comprehensive blood culture surveillance data from multiple healthcare facilities in and nearby the Kathmandu Valley.

## Methods

### Ethics statement

This study received ethical approval from institutional review boards at Stanford University, Nepal Health Research Council, and Kathmandu University School of Medical Sciences. Written informed consent was obtained from participants or their guardians prior to study enrollment. Confidentiality of participant data was ensured by de-identifying all records and aggregating blood culture results prior to analysis.

### Study design and sites

The SEAP study conducted surveillance for enteric fever at 23 hospitals and clinics in and around the Kathmandu Valley of Nepal, as previously described [8]. Sites were selected on the basis of blood culture availability and at least 20–30 cases of enteric fever per year. For this sub-study, we included four sites that participated in SEAP since its inception in September 2016 and had retrospective records of demographic and blood culture outcomes. The first site, Dhulikhel Hospital Kathmandu University Hospital, located in Kavrepalanchowk, a peri-urban area 30 km from the Kathmandu Valley, served as a prospective surveillance site. The second site, Kathmandu Model Hospital, is located in urban Kathmandu; it served as a retrospective surveillance site from September 2016 and a prospective surveillance site from October 2020

onward. For prospective surveillance, patients with fever lasting 3 or more days within the past 7 days prior to hospital presentation who resided in predefined catchment areas, or inpatients clinically suspected of enteric fever, were enrolled. In Nepal, patients typically self-present to healthcare facilities, as formal general practitioner referral services are not available in this setting. Demographic and clinical data were obtained through structured questionnaires and blood cultures were performed to identify typhoidal *Salmonella* or other pathogens. Additionally, retrospective enrollment included all *Salmonella*-positive cases identified in the hospital laboratory. The final two sites, located in the urban area of the Kathmandu Valley, are Helping Hands Hospital and Bir Hospital. These sites were retrospective surveillance sites, where individuals with culture-confirmed enteric fever were enrolled after the results of their cultures were available, typically within 1–2 weeks.

## Data collection

We collected data on all blood cultures performed and their results from October 2016 through April 2024 across four SEAP study sites. Additionally, we gathered age and gender data of all participants who underwent blood culture testing at these sites, along with their culture outcomes, starting in January 2018. The isolation and identification of *Salmonella* and other bacteria were done using standard blood culture techniques in accordance with CLSI guidelines 2020 [20]. Other bacteremia was defined as blood cultures from which one or more of the following organisms was isolated: *Escherichia coli*, *Acinetobacter* spp., *Staphylococcus aureus, Enterococcus* spp., *Klebsiella pneumoniae, Klebsiella oxytoca, Pseudomonas aeruginosa, Citrobacter* spp., *Burkholderia* spp., *Haemophilus influenzae, Morganella morganii*, *Streptococcus pyogenes, Streptococcus pneumoniae,* Viridians group *Streptococci, Proteus* spp., *Enterobacter* spp., or *Neisseria meningitidis*. Organisms such as Coagulase Negative Staphylococci (CoNS), aerobic bacilli, or *Micrococcus* spp. were considered contaminants. Daily COVID-19 case data were obtained from Our World in Data, which sources data from the World Health Organization [21]. Nepal introduced TyphiBEV (Vi-CRM197), a typhoid conjugate vaccine, into its national immunization program on April 13, 2022. Individual-level data on typhoid vaccine receipt were not available for the majority of participants in this study.

## Data analysis

We analyzed the monthly trends in blood cultures, *S.* Typhi cases, and other bacteremia from January 2018 to April 2024. Nepal implemented a first nationwide lockdown from March 24, 2020 through July 21, 2020 to contain the transmission of the SARS-CoV-2 infections and prepare the health care system to respond to the COVID-19 pandemic. Similarly, a second lockdown was imposed on April 29, 2021 until September 1, 2021, following the surge in cases during the second wave of the pandemic [22,23]. During these lockdowns, all travel was restricted, and all borders and non-essential services were closed.

We reported trends in blood cultures performed, *S.* Typhi cases, other bacteremia cases and blood culture positivity for *S.* Typhi and other bacteria across three distinct periods: before the COVID-19 pandemic (January 2018 to March 2020); during the pandemic and prior to TCV introduction (April 2020 to March 2022); post-TCV introduction (April 2022 to April 2024). Because 2018 had an unusually high number of *S.* Typhi cases, we included data from October 2016 onwards in the visualization to better reflect the baseline trend in cases; however, complete individual-level data were only available from 2018 onward, and these data were used for subsequent analyses.

We fit multivariable logistic regression models to estimate associations between study period and *S.* Typhi positivity assuming a binomial distribution. Models included fixed effects for months of blood culture collection, study site, and study period, all treated as categorical variables, with the former selected based on plausible confounding. Model selection was confirmed using Akaike Information Criterion (AIC), with the inclusion of month as a fixed effect improving model fit. We performed the analyses using all blood cultures and a sensitivity analysis among those containing a bacterial pathogen, the latter to account for possible differences in how blood cultures were utilized during the study period. Regression

analyses were performed using R (version 4.4.1) [24]. The analysis was stratified by age groups of 15 months to 15 years (age-eligible for typhoid vaccination) or 16 years and older (not eligible for vaccination), based on age at the start of the TCV introduction in April 2022.

Data used to generate the figures and conduct descriptive analyses are publicly available in the Dryad Digital Repository [25].

## Results

A total of 62,236 blood cultures were performed across four sites during the study period of January 2018 to April 2024. Of these, 31,307 (50.30%) were performed pre-pandemic (January 2018 to March 2020), 10,051 (16.15%) were performed during the pandemic (April 2020 to March 2022), and 20,878 (33.55%) were performed post-TCV introduction (April 2022 to April 2024). Among the participants, 46.30% (28,817/62,236) were female. The majority of participants (87%) were ages 16 or older (Table 1). Overall, 3.56% (2,216/62,236) cultures were positive for a pathogen. *S.* Typhi was the most common pathogen identified with 865 cases (39.03% of blood cultures with a pathogen), followed by *Acinetobacter spp* (n = 346), *Escherichia coli* (n = 326) and *S.* Paratyphi (n = 155).

Blood culture testing exhibited a seasonal pattern, with higher numbers of testing during the summer (rainy season) and lower numbers of testing during the winter. A sharp decline in the number of blood cultures performed was observed during the COVID-19 pandemic (from a mean of 1,663 per month to 670 per month across sites), followed by gradual recovery toward pre-pandemic levels (Fig 1). The trend of other bacteremia cases followed a very similar temporal pattern to blood culture testing, including a decline during the COVID-19 pandemic (from a mean of 23 per month to 10 per month across sites) and a subsequent increase. Before the pandemic, *S.* Typhi cases peaked during the summer season and declined from October to January. However, after the onset of COVID-19 pandemic, the expected peak in the summer months was absent, and case numbers remained low throughout the pandemic and continued to remain low after the introduction of the TCV vaccine, despite the gradual increase in blood culture testing (Fig 1). After TCV introduction, cases were further lower in children under 15 while they increased in those over 15 years (S1 Fig).

**Table 1. Demographic characteristics and blood culture outcomes by study period among individuals with blood cultures in Nepal (N = 62,236).**

| | Pre-pandemic N = 31,307 | Pandemic N = 10,051 | Post-TCV N = 20,878 |
|---|---|---|---|
| **Blood Culture Result** | | | |
| *Salmonella* Typhi | 662 (2.11%) | 59 (0.59%) | 144 (0.69%) |
| Other pathogenic bacteria* | 630 (2.01%) | 249 (2.48%) | 472 (2.26%) |
| **Age Group** | | | |
| 15 months to 15 years | 4,363 (13.94%) | 1,031 (10.26%) | 2,557 (12.25%) |
| 16 years or older | 26,944 (86.06%) | 9,020 (89.74%) | 18,321 (87.75%) |
| **Sex** | | | |
| Female | 14,941 (47.72%) | 4,258 (42.36%) | 9,618 (46.07%) |
| **Site** | | | |
| Bir Hospital | 9,393 (30%) | 3,453 (34.35%) | 6,811 (32.62%) |
| Dhulikhel Hospital | 11,252 (35.94%) | 3,872 (38.52%) | 6,356 (30.44%) |
| Helping Hands Hospital | 4,076 (13.02%) | 1,496 (14.88%) | 2,505 (12%) |
| Kathmandu Model Hospital | 6,585 (21.04%) | 1,230 (12.24%) | 5,206 (24.94%) |

*Other pathogenic bacteria were those determined to be probable pathogens and included *Acinetobacter spp., E. coli, S.* Paratyphi A, *Klebsiella spp., Staphylococcus aureus,* and other organisms. TCV: Typhoid conjugate vaccine, Pre-pandemic: January 2018 to March 2020, Pandemic: April 2020 to March 2022, Post-TCV: April 2022 to April 2024.

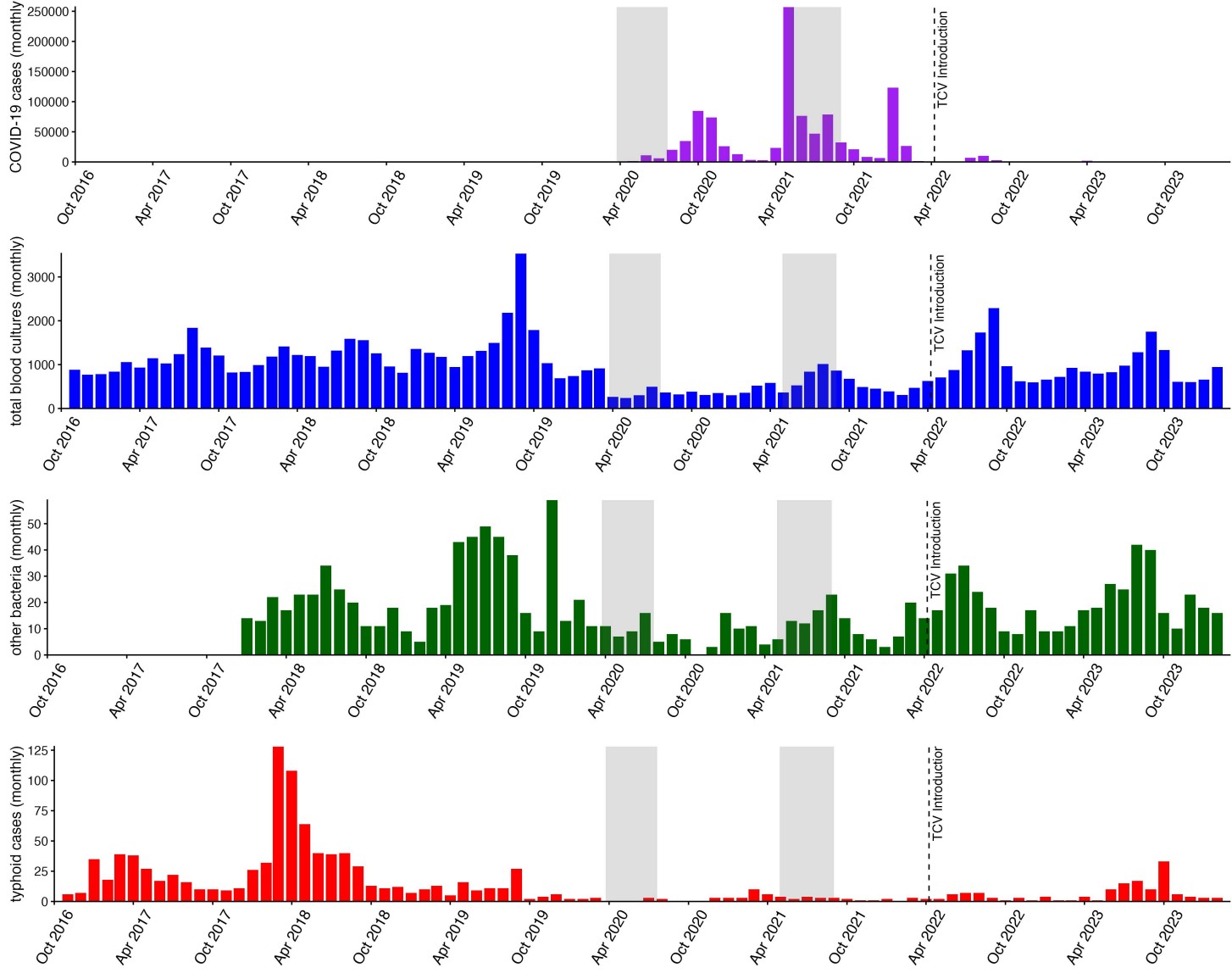

**Fig 1. Trends of monthly COVID-19 cases, blood culture testing, *Salmonella* Typhi cases, and other bacteremia cases in Nepal (2016-2024).**

The gray shaded areas reflect the time periods of the COVID-19 related lockdowns in Nepal. Purple: total COVID-19 cases, blue: total blood cultures collected, green: total other bacteremia cases, red: total other typhoid cases.

Before the pandemic, 2.11% (662/31,307) of blood cultures were positive for *S.* Typhi, which decreased to 0.59% (59/10,051) during the pandemic and continued to be low at 0.69% (144/20,878) following TCV introduction. In contrast to the decline in *S.* Typhi positivity, positivity for other pathogenic bacteria increased slightly during the pandemic (2.49%) compared to pre-pandemic (2.01%) and remained slightly higher (2.26%) during the period post-TCV introduction compared to the pre-pandemic period. *S.* Paratyphi A cases followed a similar trend to *S.* Typhi, decreasing from 0.41% (128/31,307) pre-pandemic to 0.08% (8/10,051) during the pandemic, and remained low at 0.09% (19/20,878) post-TCV introduction (Table 1).

Among children ages 15 months to 15 years, the odds of blood cultures testing positive for *S.* Typhi during the pandemic were 47% lower (adjusted odds ratio [aOR] 0.53, 95% confidence interval [CI] 0.29-0.90) than the pre-pandemic period. After TCV introduction, compared the pandemic period, the odds of culture positivity for *S.* Typhi further decreased by 75% (aOR 0.25, 95% CI 0.11-0.55). For participants 16 years and older, the odds of blood cultures testing positive for *S.* Typhi during the pandemic were 77% lower (aOR 0.22, 95% CI 0.16-0.31) than the pre-pandemic period. However, the odds of *S.* Typhi culture positivity were 59% higher (aOR 1.59, 95% CI 1.14-2.27) following TCV introduction for this age group compared to the pandemic period (Table 2).

In sensitivity analysis, when restricting the analysis to blood cultures containing any pathogen, the results were similar. The odds of blood culture positivity for *S.* Typhi were 26% lower but not statistically significant (aOR 0.74, 95% CI 0.30-1.83) in those ages 15 months to 15 years and 82% lower (aOR 0.18, 95% CI 0.12-0.26) in those 16 and older during the pandemic compared to the pre-pandemic period. After TCV introduction, compared to the pandemic period, the odds of culture positivity were 57% lower but not statistically significant (aOR 0.43, 95% CI 0.11-1.58) in those ages 15 months to 15 years and 100% higher (aOR 1.88, 95% CI 1.26-2.85) in those 16 years and older (Table 2).

## Discussion

While clinical trials demonstrated that TCVs are highly efficacious in preventing typhoid fever in those who receive them, understanding their population-level impact following introduction into national immunization programs is critical. The COVID-19 pandemic disrupted healthcare-seeking behavior, disease surveillance and possibly transmission patterns of *Salmonella* Typhi, complicating efforts to evaluate the impact of TCV introductions. Through systematic surveillance spanning the pre-pandemic, pandemic, and post-TCV introduction periods in Nepal, we found a sharp decline in the number

**Table 2. Multivariate logistic regression analysis of *Salmonella* Typhi blood culture positivity by study period and age group (Nepal).**

**Denominator: All blood cultures**

| | | aOR | 95% CI |
|---|---|---|---|
| 15m-15y | Pandemic v. Pre-pandemic | 0.53 | 0.29-0.90 |
| | Post-TCV v. Pandemic | 0.25 | 0.11-0.55 |
| ≥16y | Pandemic v. Pre-pandemic | 0.23 | 0.16-0.31 |
| | Post-TCV v. Pandemic | 1.59 | 1.14-2.27 |

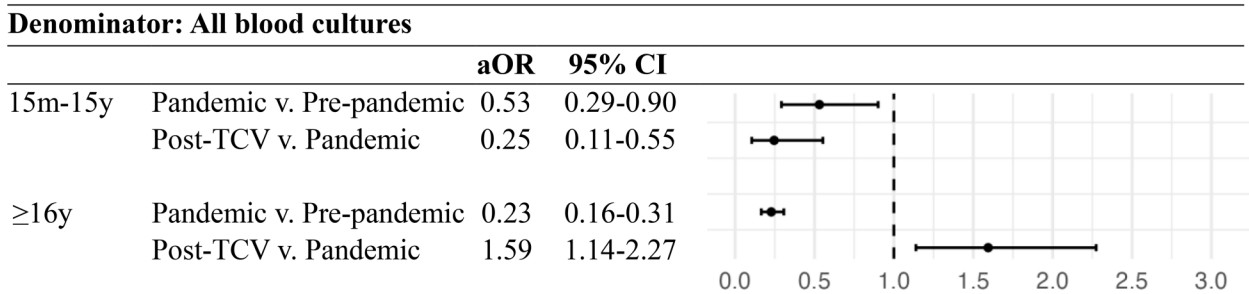

**Denominator: Blood cultures containing any pathogen**

| | | aOR | 95% CI |
|---|---|---|---|
| 15m-15y | Pandemic v. Pre-pandemic | 0.74 | 0.30-1.83 |
| | Post-TCV v. Pandemic | 0.43 | 0.11-1.58 |
| ≥16y | Pandemic v. Pre-pandemic | 0.18 | 0.12-0.26 |
| | Post-TCV v. Pandemic | 1.88 | 1.26-2.84 |

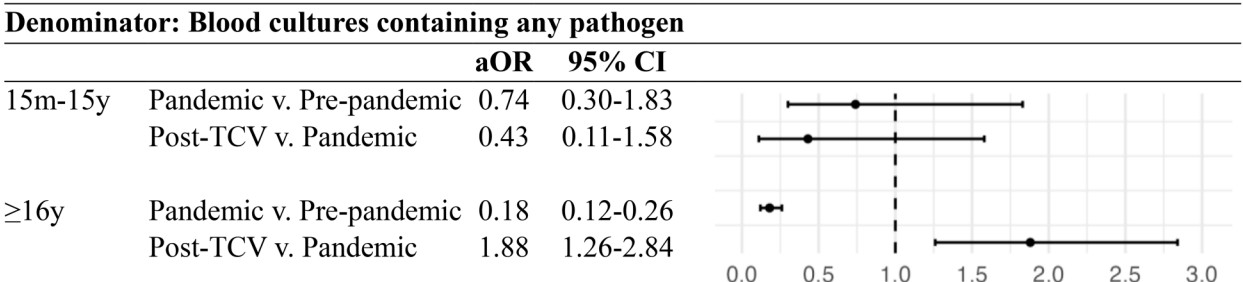

Abbreviations: aOR, Adjusted Odds Ratio (adjusted for month and study site), CI, Confidence Interval, Pre-pandemic: January 2018 to March 2020, Pandemic: April 2020 to March 2022, Post-TCV: April 2022 to April 2024, m, Months, y, Years.

of blood cultures and blood culture–confirmed typhoid cases before TCV rollout during the pandemic, likely reflecting effects of pandemic control measures on typhoid epidemiology and healthcare utilization. After introduction of vaccination, blood culture positivity for *S*. Typhi continued to significantly decrease in the younger age group, whereas in those ages 16 years and above, positivity increased after TCV rollout. Taken together, these findings indicate that TCV introduction reduced typhoid burden in the age-eligible population, but that disease burden is rebounded in older age groups following relaxation of COVID-19 measures.

The decline in the total number of blood cultures performed during the COVID-19 pandemic is likely a result of several factors, including reduced hospital visits due to travel restrictions, fear of nosocomial COVID-19 transmission, redirection of laboratory resources towards COVID-19 testing and modifications in clinical practice, such as empiric management to reduce movement of febrile patients in the hospital [26]. As the pandemic waned, the total number of blood cultures performed gradually recovered but remained below pre-pandemic levels. A similar pattern of decline during COVID-19 restrictions, followed by a return to pre-pandemic levels, were observed in Bangladesh and Pakistan [27].

The trend of blood cultures positive for bacterial pathogens other than *S*. Typhi closely mirrored the trend of blood culture testing. The identification of other bacteremia was similar in pre-COVID, during COVID and post-vaccination periods, suggesting that the reductions in blood culture utilization were not driven primarily by alterations in typhoid epidemiology, but rather in overall reductions in healthcare utilization.

*S*. Typhi positivity among blood cultures declined after the onset of the pandemic and before the introduction of vaccine in both the younger age group eligible for the TCV vaccine as well as the older age group who was not eligible. Moreover, there were larger decreases in the older age group in comparison to the younger age group. During the COVID-pandemic, improved hygiene behavior and practices may have contributed to reduced transmission of other pathogens including *S*. Typhi [28,29]. Additionally, pandemic-era social restrictions limited eating or drinking outside of the home, thereby decreasing exposure to *S*. Typhi. It is possible that older individuals were more likely to be exposed to *S*. Typhi outside the household in the pre-pandemic period, as they are more likely to eat at restaurants or consume street food, and that the reduction in this exposure caused greater reductions in typhoid risk. Similar declines in *S*. Typhi blood culture positivity during the COVID-19 pandemic were observed in Pakistan [30] and Bangladesh [24]. Reductions in reported cases of other enteric infections during COVID-19 have been observed in the United States [15], Israel [12], England [31], China [19], Spain [16], Taiwan [17], and Japan [18]. However, in Madagascar, delays in seeking care led to a surge of typhoid intestinal perforation cases [32].

In Nepal, after the introduction of TCV, typhoid positivity further decreased in the vaccine-eligible age group while it increased in the ineligible age group. While three randomized controlled trials and several observational studies have demonstrated TCVs to be highly effective at preventing typhoid fever among recipients [33], one cluster-randomized trial did not show clear evidence of indirect effects [6]. In the present study, although *S*. Typhi culture positivity declined in the vaccine eligible population following TCV introduction, it increased in older, vaccine-ineligible age groups during the post-COVID-19 period, when non-pharmaceutical pandemic interventions were scaled back and social activities started to return to pre-COVID patterns. While this pattern does not exclude the possibility that indirect effects blunted the increase in the unvaccinated individuals, it does not provide evidence for effects in this age group and does highlight the ongoing risk that this group faces in typhoid-endemic settings. Additionally, our study period ends approximately two years following TCV introduction, limiting our ability to assess longer-term vaccine effectiveness and durability of protection.

This study has several limitations. First, TCV vaccination status of the participants was not available for the vast majority of participants, precluding analysis of the direct effects of TCV on typhoid risk. This is particularly important as, to date, there are no published data on the clinical effectiveness of the TYPHIBEV (Vi-CRM197) vaccine, which was the TCV introduced in Nepal. A separate, smaller sub-study is being performed to evaluate the effectiveness of this vaccine in Nepal. Second, antibiotic use prior to presentation to the study facilities might have varied over the study periods and could have influenced blood culture sensitivity; however, we did not find differences in self-reported antibiotic utilization over the study period [27,34,35]. Third, *S*. Typhi cases were unusually high in 2018 compared with prior years, such that the decline during and

after 2020 was accentuated; nevertheless, compared with data from earlier years (Fig 1), *S*. Typhi cases were still markedly lower during the COVID-19 period. Fourth, although our study included both urban and peri-urban hospitals, the findings may not be fully generalizable to all regions of Nepal, particularly rural areas with different healthcare access and transmission dynamics. However, these sites reflect risk in and around the Kathmandu Valley and provide early evidence on typhoid trends following TCV introduction. Our analysis included sites that had both prospective enrollment of febrile patients and retrospective enrollment based on blood cultures routinely performed by the laboratory. Prospective enrollment increased the number of blood cultures performed at a site. We used fixed effects for each site to account for differences between sites and analyzed culture positivity rather than case counts, but differences over time might have still impacted the results.

Lastly, the low number of *S*. Typhi cases during the pandemic and post-TCV introduction period limit statistical power to definitively attribute the lower burden to vaccination. *S*. Paratyphi A would make for an excellent negative control outcome due to shared transmission pathways, but when stratifying by age, we had too few cases to evaluate trends by study period. Although the reduction in *S. Typhi* positivity among vaccine-eligible individuals following TCV introduction is consistent with vaccine impact, we also observed a concurrent decline in positivity for other bacteria in this age group (S1 Table). This pattern indicates that some of the observed reduction may reflect broader changes beyond TCV rollout, and we are unable to fully explain these findings. We therefore interpret the post-TCV decline with caution and acknowledge this as a major limitation of our analysis.

In conclusion, blood culture utilization and positivity for *S*. Typhi declined substantially during the COVID-19 pandemic, but *S*. Typhi positivity rebounded after 2022 in vaccine-ineligible older age groups. In contrast, blood culture positivity for *S*. Typhi further declined in vaccine-eligible age groups after the national introduction of TCV. This study reinforces the critical role of TCV in typhoid control in Nepal and provides valuable evidence to support the integration of the WHO pre-qualified typhoid vaccine in immunization programs. The persistent burden in older age groups, however, indicates need for further intervention. Comprehensive control requires sustainable investments in water and sanitation infrastructure [36], alongside consideration of expanded age eligibility for typhoid vaccination.

## Supporting information

**S1 Table. Multivariate logistic regression analysis of blood culture positivity for other pathogenic bacteria by study period and age group (Nepal).**
(DOCX)

**S1 Fig. Trends of monthly Salmonella Typhi cases in Nepal by age.** The top panel shows cases among children ages 15 months to 15 years, and the bottom panel shows cases in individuals over 15 years of age. The gray shaded areas reflect the time periods of the COVID-19 related lockdowns in Nepal.
(PNG)

## Acknowledgments

The authors thank Sudan Maharjan, Melina Thapa, Suraj Jakibanjar, Barsha Koirala, Neeru Suwal, Sarita Gosain, Mamata Maharjan, Nisha Shrestha, Natasha Shrestha, Nisha Ale, and Suvash Tamang (Dhulikhel Hospital); Himadri Neupane, Mitra Rai, and Renu Rey (Helping Hands Hospital); Archana Lamichhane and Bipin Gopali (Bir Hospital); Rishma Jal Pun and Bijay Dangol (Kathmandu Model Hospital) for their valuable support in the research. We also gratefully acknowledge all participants for their time and involvement.

## Author contributions

**Conceptualization:** Dipesh Tamrakar, Stephen P. Luby, Isaac I. Bogoch, Denise O. Garrett, Rajeev Shrestha, Jason R. Andrews.

**Data curation:** Shiva Ram Naga, Kate Doyle, Jessica C. Seidman, Alice S. Carter.

**Formal analysis:** Dipesh Tamrakar, Shiva Ram Naga, Esther Jung, Jason R. Andrews.

**Funding acquisition:** Stephen P. Luby, Denise O. Garrett.

**Investigation:** Dipesh Tamrakar, Shiva Ram Naga, Basudha Shrestha, Pratibha Bista Roka, Rabin Pokharel, Sabin Bikram Shahi, Aarjya Tara Bajracharya, Surendra K. Mahadup, Nishan Katuwal.

**Writing – original draft:** Dipesh Tamrakar, Shiva Ram Naga, Esther Jung, Jason R. Andrews.

**Writing – review & editing:** Dipesh Tamrakar, Shiva Ram Naga, Esther Jung, Basudha Shrestha, Pratibha Bista Roka, Rabin Pokharel, Sabin Bikram Shahi, Aarjya Tara Bajracharya, Surendra K. Mahadup, Nishan Katuwal, Kate Doyle, Jessica C. Seidman, Alice S. Carter, Stephen P. Luby, Isaac I. Bogoch, Kristen Aiemjoy, Denise O. Garrett, Rajeev Shrestha, Jason R. Andrews.

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
