## [Decision Letter · Decision Letter 0]

30 Sep 2025

Impact of the COVID-19 pandemic and typhoid conjugate vaccine introduction on typhoid fever in Nepal

Dear Dr. Andrews,

Thank you for submitting your manuscript to PLOS Neglected Tropical Diseases. After careful consideration, we feel that it has merit but does not fully meet PLOS Neglected Tropical Diseases's publication criteria as it currently stands. Therefore, we invite you to submit a revised version of the manuscript that addresses the points raised during the review process.

Please submit your revised manuscript within 60 days Nov 29 2025 11:59PM. If you will need more time than this to complete your revisions, please reply to this message or contact the journal office at plosntds@plos.org. Please include the following items when submitting your revised manuscript:

We look forward to receiving your revised manuscript.

Kind regards,

Richard A. Bowen, DVM PhD

Academic Editor

Stuart Blacksell

Section Editor

Shaden Kamhawi

co-Editor-in-Chief

Paul Brindley

co-Editor-in-Chief

**Additional Editor Comments:**

Your manuscript has been reviewed and we request that you evaluate and respond to the comments made by reviewers, modify the manuscript accordingly and resubmit.

**Journal Requirements:**

At this stage, the following Authors/Authors require contributions: Dipesh Tamrakar, Shiva Ram Naga, Esther Jung, Basudha Shrestha, Pratibha Bista Roka, Rabin Pokharel, Sabin Bikram Shah, Aarjya Tara Bajracharya, Surendra Mahadup, Nishan Katuwal, Kate Doyle, Jessica Couvillion Seidman, Alice Carter, Stephen Luby, Isaac I. Bogoch, Kristen Aiemjoy, Denise Garrett, Rajeev Shrestha, and Jason Andrews. Please ensure that the full contributions of each author are acknowledged in the "Add/Edit/Remove Authors" section of our submission form.

Potential Copyright Issues:

- Figure 1. Please (a) provide a direct link to the base layer of the map (i.e., the country or region border shape) and ensure this is also included in the figure legend; and (b) provide a link to the terms of use / license information for the base layer image or shapefile. We cannot publish proprietary or copyrighted maps (e.g. Google Maps, Mapquest) and the terms of use for your map base layer must be compatible with our CC BY 4.0 license.

5) Please ensure that the funders and grant numbers match between the Financial Disclosure field and the Funding Information tab in your submission form. Note that the funders must be provided in the same order in both places as well.

State the initials, alongside each funding source, of each author to receive each grant. For example: "This work was supported by the National Institutes of Health (####### to AM; ###### to CJ) and the National Science Foundation (###### to AM).".

**Reviewers' Comments:**

Reviewer's Responses to Questions

**Key Review Criteria Required for Acceptance?**

**Methods:**

-Are the objectives of the study clearly articulated with a clear testable hypothesis stated?

-Is the study design appropriate to address the stated objectives?

-Is the population clearly described and appropriate for the hypothesis being tested?

-Is the sample size sufficient to ensure adequate power to address the hypothesis being tested?

-Were correct statistical analysis used to support conclusions?

-Are there concerns about ethical or regulatory requirements being met?

Reviewer #1: - Further explanation on why those sites were chosen is required. This could have attracted a selection bias, in terms of better infrastructure, better adherence, …etc.

- Could you comment on the process of presenting to healthcare facilities- like GP referral or self-presentation.

- More information on the regression analysis is required. How did you handle the variables in the regression analyses? How did you choose the final model? How did you make sure that it is the best fitting model?

- More information on how you safeguarded the confidentiality of the participants is good to be added to the ethical statement. Things like deidentification of data…etc

Reviewer #2: • How were the 4 sites chosen for this study? How representative are they of the general population? Three of the sites appear to be very close together and may have overlapping catchment areas, while one is several [districts?] away and may have a very different population.

• Please mention which TCV was used in Nepal in the Methods.

• Can the authors include the exact date of TCV introduction in this area in the Methods? Also, why is April 2022 part of the pandemic (“prior to TCV introduction” Lines 170-171) period when it was when the vaccine was introduced?

• The authors could consider doing a sensitivity analysis with pre-pandemic period excluding 2018.

• How long was the catchup campaign? And how long did it take to implement? It may not make sense to use the beginning of some of the post-pandemic vaccination period.

• Did everyone consent to enrolment? (what was the participation rate)

• The authors could consider including study site as a random effect and then looking at the within and between study site effects (in the logistic regression).

**Results:**

-Does the analysis presented match the analysis plan?

-Are the results clearly and completely presented?

-Are the figures (Tables, Images) of sufficient quality for clarity?

Reviewer #1: - Most blood cultures were obviously pre-COVID. Could this has influenced your data?

- You need to highlight that some values have wide CI and not significant, putting uncertainties on the value of the results. Example: “After TCV introduction, compared to the pandemic period, the odds of culture 252 positivity were 71% lower (aOR 0.29, 95% CI 0.05-1.35) in those ages 15 months to 15 years”

Reviewer #2: • Fig 2: It might make sense to convert the COVID-19 cases to monthly instead of daily, to match the other plots. Can the authors add what the different colors represent in the caption? Also, it would be really helpful to see all of the time series (especially the last one, typhoid cases) by study site. Can the authors also show a plot with a smaller y-axis (i.e., excluding 2018) so we can see the lower case count trends more clearly? As it is, it is very difficult to see the case numbers and patterns after 2020.

• It almost looks like cases increased after TCV introduction (by looking at Fig 2). It might be more helpful to also look at impact by year (even within each pre/during/post pandemic period) to see if it is stable within each period. Could do something like a synthetic control using other pathogens if they had a similar pre-intervention trends.

• It would be helpful to see the results for other pathogens (like Table 2 results) over the same time periods and in the same groups for comparison.

• What were the estimates for the adjustment factors in the logistic regression? Were there any substantial differences between sites (and other variables)? What were those impacts and what did they look like? Again, the authors could consider a random effect for study site or something similar.

**Conclusions:**

-Are the conclusions supported by the data presented?

-Are the limitations of analysis clearly described?

-Do the authors discuss how these data can be helpful to advance our understanding of the topic under study?

-Is public health relevance addressed?

Reviewer #1: - I have concerns about the generalisability of the findings and particularly I wouldn’t draw a broader conclusion about the impact of the TCV role out, but rather these findings reflect theses areas and only examined population-

Reviewer #2: • Some of the implications stated in the discussion are too strong, given the limitations of the methods chosen and challenges in evaluating post-pandemic data. Since the pandemic and post-pandemic cases remain low, the authors cannot state that the lower cases post-pandemic were only because of vaccination. This is a limitation of the data, but it would be possible to use other pathogens (or other non-intervention sites) over the same time period. This additional analysis is not mandatory, but these limitations need to be acknowledged further and the conclusions need to be softened if no additional analyses are included.

• Consider citing these studies in the limitations where the authors mention antibiotic use and its influence on blood culture sensitivity:

o Antillon M, Saad NJ, Baker S, Pollard AJ, Pitzer VE. The relationship between blood sample volume and diagnostic sensitivity of blood culture for typhoid and paratyphoid fever: a systematic review and meta‐analysis. J Infect Dis. 2018;218(suppl_4):S255‐S267. 10.1093/infdis/jiy471

o Phillips MT, Meiring JE, Voysey M, Warren JL, Baker S, Basnyat B, Clemens JD, Dolecek C, Dunstan SJ, Dougan G, Gordon MA, Thindwa D, Heyderman RS, Holt KE, Qadri F, Pollard AJ, Pitzer VE; STRATAA Study Group. A Bayesian approach for estimating typhoid fever incidence from large-scale facility-based passive surveillance data. Stat Med. 2021 Nov 20;40(26):5853-5870. doi: 10.1002/sim.9159. Epub 2021 Aug 24. PMID: 34428309; PMCID: PMC9291985.

These indicate that antibiotic use can greatly impact blood culture sensitivity and reporting. In Figure S2 of the second citation, not that Nepal has some of the highest self-reported antibiotic use.

• Consider exploring or commenting on how reporting might have changed through the different periods, and how it might vary by study sites. This may impact the findings.

• Similar to the previous comment, are there differences between the reporting (or underreporting) of prospective and retrospective sites/time periods?

• How do these results compare to previous modeling studies about vaccine implementation in similar settings (like this: Joke Bilcke, Marina Antillón, Zoë Pieters, Elise Kuylen, Linda Abboud, Kathleen M Neuzil, Andrew J Pollard, A David Paltiel, Virginia E Pitzer. Cost-effectiveness of routine and campaign use of typhoid Vi-conjugate vaccine in Gavi-eligible countries: a modelling study. The Lancet Infectious Diseases. Volume 19, Issue 7, 2019, Pages 728-739. https://doi.org/10.1016/S1473-3099(18)30804-1)?

• Can the authors add some more sentences about the broader public health implications of this study? For example, they mention the need for additional control measures like improved water and sanitation infrastructure in the abstract and hint at it in the introduction, but it isn’t in the discussion. Could cite this paper: Phillips MT, Owers KA, Grenfell BT, Pitzer VE (2021). Changes in historical typhoid transmission across 16 U.S. cities, 1889–1931: Quantifying the impact of investments in water and sewer infrastructures. PLOS Neglected Tropical Diseases 15(4): e0009347. https://doi.org/10.1371/journal.pntd.0009347 .

• In the limitations, the authors could mention something about multidrug resistant strains of typhoid (optional).

• Can the authors state something about the long-term effects of the vaccine that aren’t observed yet in this study (because it only spans ~2 years post vaccination)?

**Editorial and Data Presentation Modifications?**

Reviewer #1: (No Response)

Reviewer #2: Introduction:

• Consider citing

o Antillon M, Warren J, Crawford F, Weinberger D, Kurum E, Pitzer V. The burden of typhoid fever in low- and middle-income countries: a meta-regression approach. PLoS Negl Trop Dis. 2017. https://doi.org/10.1371/journal.pntd.0005376.

o GBD 2017 Risk Factor Collaborators. Global, regional, and national comparative risk assessment of 84 behavioural, environmental and occupational, and metabolic risks or clusters of risks for 195 countries and territories, 1990–2017: a systematic analysis for the Global Burden of Disease Study 2017. Lancet (London, England). 2018;392(10159):1923–94.

o Mogasale V, Maskery B, Ochiai RL, et al. Burden of typhoid fever in low‐income and middle‐income countries: a systematic, literature‐based update with risk‐factor adjustment—the lancet global health. Lancet. 2014;2(10):e570‐e580. 10.1016/S2214-109X(14)70301-8

for annual burden estimate citations instead of the current citation for a single year (2021).

• Please also consider citing:

o Mila Shakya, Merryn Voysey, Katherine Theiss-Nyland, Rachel Colin-Jones, Dikshya Pant, Anup Adhikari, Susan Tonks, Yama F Mujadidi, Peter O’Reilly, Olga Mazur, Sarah Kelly, Xinxue Liu, Archana Maharjan, Ashata Dahal, Naheeda Haque, Anisha Pradhan, Suchita Shrestha, Manij Joshi, Nicola Smith, Jennifer Hill, Jenny Clarke, Lisa Stockdale, Elizabeth Jones, Timothy Lubinda, Binod Bajracharya, Sabina Dongol, Abhilasha Karkey, Stephen Baker, Gordan Dougan, Virginia E Pitzer, Kathleen M Neuzil, Shrijana Shrestha, Buddha Basnyat, Andrew J Pollard. Efficacy of typhoid conjugate vaccine in Nepal: final results of a phase 3, randomised, controlled trial. The Lancet Global Health. Volume 9, Issue 11. 2021. Pages e1561-e1568. https://doi.org/10.1016/S2214-109X(21)00346-6.

for efficacy of Phase 3 trial of TCVs in Nepal. There are also several others, but please at least include this one.

Minor comments:

• Please spell out acronyms (like aOR and CI) before using them the first time.

• Please be consistent with commas and decimal points in numbers (sometimes there are commas in the larger numbers and sometimes there aren’t, and numbers after the decimals vary from 0-2 throughout the text and tables).

• Please provide more detailed table and figure titles and captions. The titles should be able to stand alone—please give study setting and dates.

• The link provided for data does not work, and it is unknown if the data are there

**Summary and General Comments:**

Reviewer #1: A well-written manuscript, few recommendations to improve it.

Reviewer #2: In “Impact of the COVID-19 pandemic and typhoid conjugate vaccine introduction on typhoid fever in Nepal,” the authors investigated the impact of TCV introduction after the COVID pandemic in Nepal. This is a very timely study with interesting implications, and can fill some much-needed gaps about routine use of typhoid conjugate vaccines. That being said, some of the implications stated in the discussion are too strong given the limitations of the methods chosen and challenges in evaluating post-pandemic data. Based on these methods and data, it is not possible to say that the decline in typhoid is purely because of TCVs. I would suggest either softening some of these conclusions, or doing some additional or alternative analyses to be able to say this. I would also recommend doing a literature search on this topic before writing the manuscript in the future, as some of the citations could be updated. Please see attached document for all comments.

PLOS authors have the option to publish the peer review history of their article (what does this mean? ). If published, this will include your full peer review and any attached files.

**Do you want your identity to be public for this peer review?** For information about this choice, including consent withdrawal, please see our Privacy Policy .

Reviewer #1: No

Reviewer #2: No

**Figure resubmission:**
---

## [Decision Letter · Decision Letter 1]

18 Dec 2025

Dear Dr. Andrews,

We are pleased to inform you that your manuscript 'Impact of the COVID-19 pandemic and typhoid conjugate vaccine introduction on typhoid fever in Nepal' has been provisionally accepted for publication in PLOS Neglected Tropical Diseases.

Best regards,

Richard A. Bowen, DVM PhD

Academic Editor

Stuart Blacksell

Section Editor

Shaden Kamhawi

co-Editor-in-Chief

Paul Brindley

co-Editor-in-Chief

Thank you for the thoughtful and careful response to reviewer comments.

Reviewer's Responses to Questions

**Key Review Criteria Required for Acceptance?**

**Methods**

-Are the objectives of the study clearly articulated with a clear testable hypothesis stated?

-Is the study design appropriate to address the stated objectives?

-Is the population clearly described and appropriate for the hypothesis being tested?

-Is the sample size sufficient to ensure adequate power to address the hypothesis being tested?

-Were correct statistical analysis used to support conclusions?

-Are there concerns about ethical or regulatory requirements being met?

Reviewer #2: (No Response)

Reviewer #3: The retrospective design is perfectly fitted for this research.....however, applying prospective design also in one of the cetres would try to invalidate the results.

**Results**

-Does the analysis presented match the analysis plan?

-Are the results clearly and completely presented?

-Are the figures (Tables, Images) of sufficient quality for clarity?

Reviewer #2: (No Response)

Reviewer #3: Good enough

**Conclusions**

-Are the conclusions supported by the data presented?

-Are the limitations of analysis clearly described?

-Do the authors discuss how these data can be helpful to advance our understanding of the topic under study?

-Is public health relevance addressed?

Reviewer #2: (No Response)

Reviewer #3: Appropriate

**Editorial and Data Presentation Modifications?**

Reviewer #2: (No Response)

Reviewer #3: All well

**Summary and General Comments**

Reviewer #2: No further comments. The authors have adequately responded to my suggestions. As mentioned before, this manuscript is very timely and can have important implications for TCVs and typhoid control.

Reviewer #3: Just a cliche with the study design, rest of the things appear good.

PLOS authors have the option to publish the peer review history of their article (what does this mean? ). If published, this will include your full peer review and any attached files.

**Do you want your identity to be public for this peer review?** For information about this choice, including consent withdrawal, please see our Privacy Policy .

Reviewer #2: No

Reviewer #3: **Yes:** Bikash Shrestha

---

## [Editor Report · Acceptance letter]

Dear Dr. Andrews,

We are delighted to inform you that your manuscript, "Impact of the COVID-19 pandemic and typhoid conjugate vaccine introduction on typhoid fever in Nepal," has been formally accepted for publication in PLOS Neglected Tropical Diseases.

Best regards,

Shaden Kamhawi

co-Editor-in-Chief

Paul Brindley

co-Editor-in-Chief
